# Can You Win Everything with A Lottery Ticket?

**Tianlong Chen**                                                   *tianlong.chen@utexas.edu*
*University of Texas at Austin*

**Zhenyu Zhang**                                                    *zhenyu.zhang@utexas.edu*
*University of Texas at Austin*

**Jun Wu**                                                          *jwum@amazon.com*
*Amazon Web Services*

**Randy Huang**                                                     *renfu@amazon.com*
*Amazon Web Services*

**Sijia Liu**                                                       *liusiji5@msu.edu*
*Michigan State University*
*MIT-IBM Watson AI Lab, IBM Research*

**Shiyu Chang**                                                     *chang87@ucsb.edu*
*University of California, Santa Barbara*

**Zhangyang Wang**                                                  *atlaswang@utexas.edu*
*University of Texas at Austin*

**Reviewed on OpenReview:** *https://openreview.net/forum?id=JL6MU9XFzW*

## Abstract

*Lottery ticket hypothesis* (LTH) has demonstrated to yield independently trainable and highly sparse neural networks (a.k.a. *winning tickets*), whose test set accuracies can be surprisingly on par or even better than dense models. However, accuracy is far from the only evaluation metric, and perhaps not always the most important one. Hence it might be myopic to conclude that a sparse subnetwork can replace its dense counterpart, even if the accuracy is preserved. Spurred by that, we perform the first comprehensive assessment of lottery tickets from diverse aspects beyond test accuracy, including *(i)* generalization to distribution shifts, *(ii)* prediction uncertainty, *(iii)* interpretability, and *(iv)* geometry of loss landscapes. With extensive experiments across datasets {CIFAR-10, CIFAR-100, and ImageNet}, model architectures, as well as seven sparsification methods, we thoroughly characterize the trade-off between model sparsity and the all-dimension model capabilities. We find that an appropriate sparsity (e.g., $20\% \sim 99.53\%$) can yield the winning ticket to perform comparably or even better **in all above four aspects**, although some aspects (generalization to certain distribution shifts, and uncertainty) appear more sensitive to the sparsification than others. We term it as a `LTH-PASS`. Overall, our results endorse choosing a good sparse subnetwork of a larger dense model, over directly training a small dense model of similar parameter counts. We hope that our study can offer more in-depth insights on pruning, for researchers and engineers who seek to incorporate sparse neural networks for user-facing deployments. Codes are available in `https://github.com/VITA-Group/LTH-Pass`.

## 1 Introduction

State-of-the-art pruning techniques are able to remove the majority of weights from deep neural networks (DNNs) almost without compromising the test accuracy (Mozer & Smolensky, 1989; Janowsky, 1989; LeCun

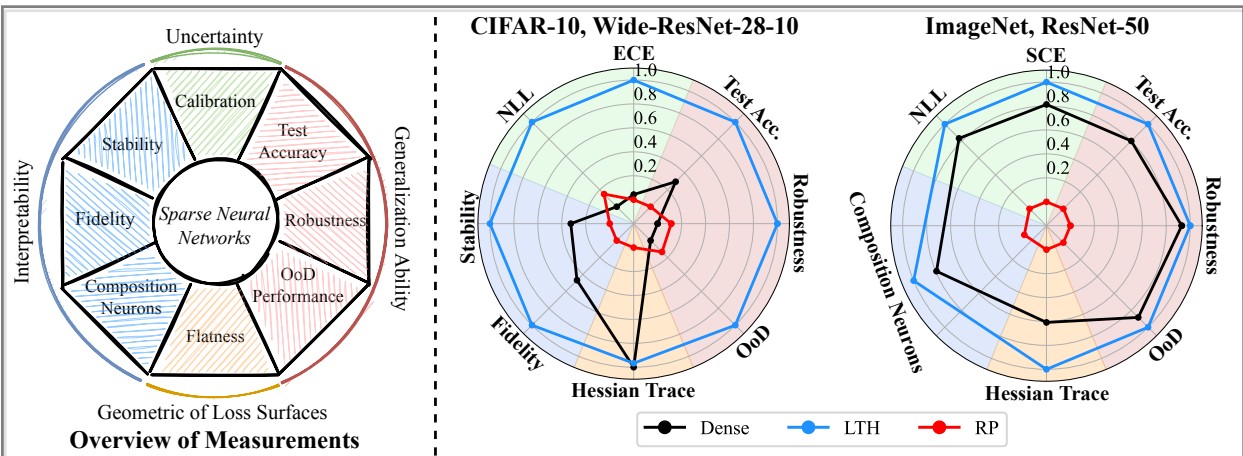

Figure 1: Overall assessments of sparse neural networks. *Left* figure presents an overview of measurements. *Right* figures show achieved full-scale performance, where **outer cycles indicate superior (sparse) networks**. **Dense** is the unpruned full model; LTH (Frankle & Carbin, 2019) denotes the winning ticket identified from the dense network, which is also a `LTH-PASS` here; RP represents a randomly pruned sparse model. All sparse networks on CIFAR/ImageNet only have 32.77%/51.20% parameters of the dense model. Associated with the *left* figure, {Test Accuracy, Robustness, OoD}, {Hessian Trace}, {Fidelity, Stability, Composition Neurons}, {NLL, ECE or SCE} are measurements for generalization ability, geometric of loss surfaces, interpretability, uncertainty respectively. Note that reported numbers of each metric are normalized by subtracting the minimum and dividing the gap between maximum and minimum values.

et al., 1990b; Han et al., 2016; Guo et al., 2016; Molchanov et al., 2016). The emerging lottery ticket hypothesis (LTH) (Frankle & Carbin, 2019) advocates that dense models contain highly sparse subnetworks, i.e., *winning tickets*, with the same good trainability, expressiveness, and transferability (Morcos et al., 2019a; Chen et al., 2020b;a) compared to their dense counterpart. All these intriguing attributes together with the remarkable efficiency lead to a wide deployment of sparse networks in a resource-constrained real world (Lane & Warden, 2018). However, while many works narrowly refer to model "performance" as its test set accuracy, researchers have been long aware of the more complicated myriad of performance dimensions. Indeed, it remains elusive whether or not there are hidden pitfalls in a winning lottery ticket, besides the test accuracy versus efficiency, i.e., *have we missed or overlooked any unexpected loss along other performance dimensions when we prune a neural network?* This is the central question motivating the current work.

There were preliminary attempts done in earlier literature (Hooker et al., 2019; 2020b; Gui et al., 2019; Ye et al., 2019; Wang et al., 2018; Zhou et al., 2009; Venkatesh et al., 2020; Chen et al., 2020b;a; Koohpayegani et al., 2020; Morcos et al., 2019b; Zhang et al., 2021a; Sakamoto & Sato, 2022; Chen et al., 2022c) trying to address some part of this question. Some researchers advocated the existence of sparse subnetworks (winning tickets) with comparable transferability to the full dense models (Chen et al., 2020b;a; Koohpayegani et al., 2020; Morcos et al., 2019b; Chen et al., 2022a) and adversarial robustness (Chen et al., 2022b; Gui et al., 2019; Ye et al., 2019). Other recent works (Hooker et al., 2019; 2020b) pointed out that sparse networks are brittle to small changes such as natural image corruptions, and might amplify the class imbalance more than dense counterparts (Hooker et al., 2020a). Many other important aspects, such as uncertainty, interpretability ,and loss landscape, are not well studied as performance criteria in sparse neural networks, up to our best knowledge. In many applications, we cannot afford to pay any of them as the "hidden price" of sparsification. For example, robustness and interpretability are stipulated by safety-critical scenarios like autonomous cars and medical diagnostic, respectively. With those aspects under-scrutinized, it is hard to draw decisive conclusions on whether sparse winning tickets can become a drop-in replacement for dense ones, despite their appealing accuracy-efficiency trade-offs, and putting off their wider adoption with many open concerns. A comprehensive study into answering this question is thus highly demanded.

To the best of our knowledge, our work for the first time systematically characterizes and quantifies the full-dimension performance of sparse neural networks obtained by LTH and other pruning mechanisms. Specifically, we assess sparsity from its impacts on four carefully picked perspectives (Figure 1), including

*generalization* to distribution shifts, *uncertainty* quantification, *interpretability*, and *loss geometry* that locally assess the learned functions. When an identified sparse subnetwork can be separately trained, to match all the above four aspects as the full dense model can do - we name it a `LTH-PASS`. Our contributions are outlined:

⋆ We define a more rigorous notion, `LTH-PASS`, which requires located subnetworks to match **all measured aspects** with their dense networks, i.e. with the same or even better ability to generalize to various shifted data distributions, to quantify uncertainty, to be interpreted (especially by neuron explanations), and to preserve learned functional approximation as indicated by loss landscape geometry, in addition to the unimpaired test accuracy. Such `LTH-PASS` is identified at $20\% \sim 99.53\%$ sparsity from diverse scenarios.

⋆ The excessive sparsification can often deteriorate any of the above aspects, and some aspects are far more sensitive to sparsity than others, e.g., the generalization to distribution shifts (e.g., with natural corruptions or adversarial perturbations) and the uncertainty quantification.

⋆ We also observe that it is advantageous to choose a sparse subnetwork (i.e., `LTH-PASS`) of a larger dense model, than directly using a small dense network with similar parameter counts (Figure 11), along all those performance dimensions. That implies the role of sparsity as a sophisticated, comprehensive regularization affecting multiple aspects of neural networks, rather than just ad-hoc reduction of parameters.

⋆ The above insights are drawn from extensive experiments across multiple datasets: CIFAR-10, CIFAR-100 (Krizhevsky & Hinton, 2009) and ImageNet (Deng et al., 2009); using diverse dense model architectures: ResNets (He et al., 2016) and Wide-ResNets (Zagoruyko & Komodakis, 2016); as well as performing seven representative sparsification regimes such as magnitude pruning (Han et al., 2016), lottery ticket hypothesis (Frankle & Carbin, 2019), random pruning, pruning at initialization (Lee et al., 2019; Wang et al., 2020; Tanaka et al., 2020), and dynamic sparse training (Evci et al., 2020; Liu et al., 2021b). We hope our benchmarking efforts to motivate more future studies as a common ground for comparison.

## 2 Related Works

### 2.1 Pruning, Lottery Ticket Hypothesis, and Dynamic Sparsity

Weight pruning can effectively eliminate redundancy in deep neural networks (LeCun et al., 1990b; Han et al., 2016) and obtain storage and computational savings. In general, it contains the following iterative cycles: *(a)* training the dense neural networks for at least several iterations; *(b)* removing unnecessary weights according to certain criteria and deriving subnetworks; *(c)* fine-tuning obtained sparse model to recover accuracy. Different sparsity patterns may be pursued, from unstructured (Han et al., 2015; LeCun et al., 1990a; Han et al., 2016) to structured sparsity (Liu et al., 2017; He et al., 2017; Zhou et al., 2016), the former being more flexible while the latter is often treated as more hardware friendly.

One of the mainstream pruning techniques is magnitude-based, which zeroes out a percentage of model weights by thresholding their magnitudes (Han et al., 2015; 2016). Later methods (Blalock et al., 2020) perform thresholding based on gradients (Molchanov et al., 2016; 2019) or hessian (LeCun et al., 1990a; Hassibi & Stork, 1992; Hassibi et al., 1993) based measures, instead of raw element magnitudes. The iterative pruning fashion (Han et al., 2016; Zhu & Gupta, 2017; Tan & Motani, 2020; Liu et al., 2019c) is often adopted for ameliorating performance degradation. Other pruning strategies formulate pruning as optimization objectives, by incorporating sparsity-promoting regularization (Liu et al., 2017; He et al., 2017; Zhou et al., 2016) or by constrained optimization (Boyd et al., 2011; Ouyang et al., 2013; He et al., 2017; Luo et al., 2017; Yu et al., 2018; Aghasi et al., 2017; Serra et al., 2020; ElAraby et al., 2020; Serra et al., 2021).

Lottery ticket hypothesis (LTH) (Frankle et al., 2019) recently emerges to investigate the independent trainable, extremely sparse neural networks from scratch, which are capable of recovering or even surpassing the original dense network's performance. Sahu et al. (2022) points out that smaller models benefit more from the ticket search. To scale up LTH for large networks and large-scale datasets, weight rewinding techniques (Renda et al., 2020; Frankle et al., 2020a) is proposed. The intriguing properties of LTH received wide attention and have been broadly explored in various contexts, such as image classification (Frankle & Carbin, 2019; Liu et al., 2019b; Wang et al., 2020; Evci et al., 2019; Frankle et al., 2020b; Savarese

Table 1: Details of training configurations for experiments with OMP, LTH, RP, PI approaches.

| Dataset | Learning Rate | Batch Size | Epochs | Optimizer | Momentum | Weight Decay |
|---|---|---|---|---|---|---|
| CIFAR-10/100 | 0.1; ×0.1 at 91,136 epoch | 128 | 182 | SGD | 0.9 | $1 \times 10^{-4}$ |
| ImageNet | 0.4; ×0.1 at 30,60,80 epoch; linearly warmup 5 epochs | 1024 | 90 | SGD | 0.9 | $1 \times 10^{-4}$ |

et al., 2020; You et al., 2020; Chen et al., 2020a; 2022c;a), object detection (Girish et al., 2020), natural language processing (Gale et al., 2019; Yu et al., 2020; Prasanna et al., 2020; Chen et al., 2020b;c), generative adversarial networks (Kalibhat et al., 2020; Chen et al., 2021a), graph neural networks (Chen et al., 2021b), reinforcement learning (Yu et al., 2020), and lifelong learning (Chen et al., 2021c). Most of them leverage unstructured iterative magnitude pruning (Han et al., 2016; Frankle & Carbin, 2019) to identify the *winning tickets*, which we also follow in this work.

To save resources at training stages, SNIP (Lee et al., 2019), GraSP (Wang et al., 2020), and SynFlow (Tanaka et al., 2020) can be introduced to obtain high-quality sparse subnetworks before the training process starts, i.e., at the random initialization, based on several salience criteria. Another related field is dynamic sparse training (DST) (Mocanu et al., 2018; Liu et al., 2020) which trains sparse neural networks from scratch by optimizing the sparse connectivity and model parameters simultaneously. Numerous approaches (Mocanu et al., 2016; Evci et al., 2019; Mostafa & Wang, 2019; Dettmers & Zettlemoyer, 2019; Liu et al., 2021a; Dettmers & Zettlemoyer, 2019; Evci et al., 2020; Jayakumar et al., 2020; Raihan & Aamodt, 2020; Liu et al., 2021b) study such dynamic sparsity, often matching state-of-the-art training performance (Liu et al., 2021b).

## 2.2 Measurements of Sparse Neural Networks

Although the test set accuracy is often the core interest, more researchers start to examine and characterize the impact of pruning from more perspectives beyond that. (1) Compression w.r.t. *fairness*: (Hooker et al., 2019; 2020b; Paganini, 2020) demonstrates compression may amplify existing algorithmic bias on the underrepresented long-tail of the data distribution, which is at odds with fairness objectives, and potentially results in disparate treatments of protected attributes (Zink & Rose, 2020). (2) Compression w.r.t. *robustness*: (Gui et al., 2019; Ye et al., 2019) show that with an appropriate sparsity, pruned subnetworks are capable of maintaining unimpaired adversarial robustness and standard accuracy. (Hooker et al., 2019; 2020b) tell a different story that compressed models are more sensitive and brittle to shifted data distributions such as natural corrupted samples (Hendrycks & Dietterich, 2019). (3) Compression w.r.t. *privacy*: (Wang et al., 2018; Zhou et al., 2009; Gondara et al., 2021) enable sparse models to obtain a strong differential-privacy guarantee. (4) Compression w.r.t. *transferability*: extensive investigations (Chen et al., 2020b;a; Koohpayegani et al., 2020; Morcos et al., 2019b; Iofinova et al., 2022) indicate that there exist high quality subnetworks with competitive or even enhanced transferability across diverse datasets. (5) Compression w.r.t. *uncertainty*: (Venkatesh et al., 2020) integrates a suite of calibration strategies into existing pruning procedures, and locates reliable subnetworks with improved uncertainty.

## 3 Preliminary

**Network.** We use the official ResNet-20s (R20s), ResNet-18 (R18), ResNet-50 (R50) (He et al., 2016), and Wide-ResNet-28-10 (WR28-10) (Zagoruyko & Komodakis, 2016) as the original (unpruned) dense networks. $f(x; \theta)$ represents the output of a model with parameters $\theta \in \mathbb{R}^d$ and on input images $x$. Similarly, subnetworks extracted from the dense model $\theta$ can be depicted as $m \odot \theta$, where $m \in \{0, 1\}^d$ is a pruning binary mask and $\odot$ denotes the element-wise product. Note that pruning is mainly conducted over networks without counting their classification heads. Sparsity of a compressed neural network is defined as the ratio of removed parameters to the total parameters, e.g., sparsity = $1 - \frac{\|m\|_0}{d}$ in our case.

**Pruning Methods.** To find the subnetworks $m \odot \theta$, we leverage several classical pruning approaches: (1) *one-shot magnitude pruning* (OMP) by removing a portion of weights with the globally smallest magnitudes (Han et al., 2016); (2) *the lottery ticket hypothesis* (LTH) (Frankle & Carbin, 2019) with iterative weight magnitude pruning (IMP) (Han et al., 2016). Following the LTH's standard routines, it iteratively prunes the 20% of remaining weight with the globally smallest magnitudes and rewinds model weights to the same random

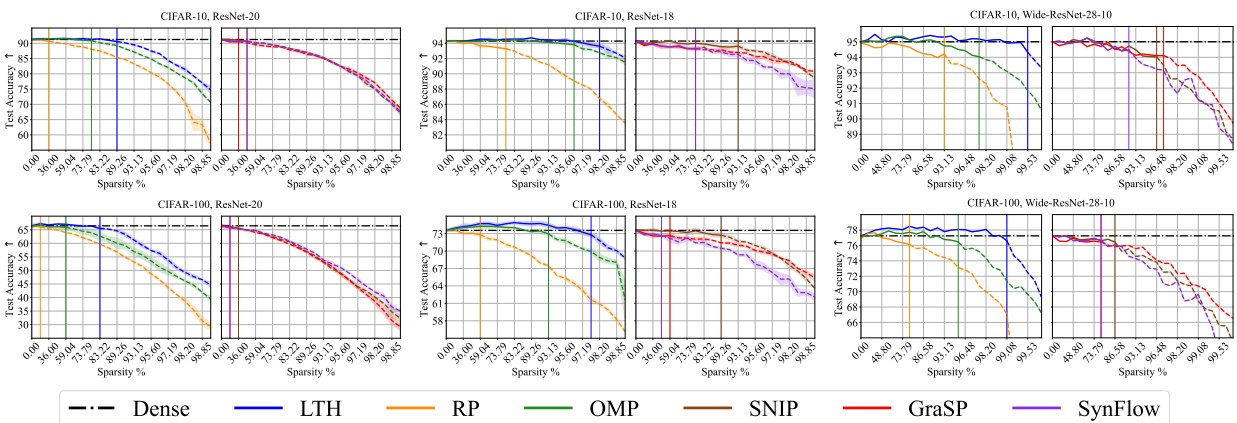

Figure 2: **Test accuracy** ($\uparrow$) of diverse subnetworks with a range of sparsity from 0.00% to 99.53% on CIFAR datasets. The sparsity levels in X-axis are obtained from iteratively pruning with a ratio of 20%, i.e., $(1 - 0.8^n) \times 100\%$ and $n$ is the number of pruning rounds. **Dense** (balck dashed lines) denotes the unpruned dense models. $\uparrow/\downarrow$ **indicate a better model should have a lager/smaller measurement**. Curves with errors (shadow regions) are the average across three independent runs, with standard deviations: same hereinafter. Each curve is divided into the region I (solid lines) of winning tickets; the region II of degraded subnetworks marked by dash lines. Regions I and II are separated by the extreme sparsity defined as the maximum sparsity when the subnetwork is at most 1% test accuracy drop compared to its dense counterpart. More detailed can be found in Table A3. The majority of our investigations are conducted on the high quality winning tickets from region I.

initialization (Frankle & Carbin, 2019) or early training epochs (Frankle et al., 2020b; Chen et al., 2020a). In our case, weights are rewound to the $3_{\mathrm{rd}}/15_{\mathrm{th}}$ epoch for CIFAR and ImageNet experiments respectively, following the setups in Chen et al. (2020a); (3) *random pruning* (RP) which usually serves as a necessary baseline for the sanity check (Frankle & Carbin, 2019); (4) *pruning at initialization* (PI) mechanisms. Some representative methods, SNIP (Lee et al., 2019), GraSP (Wang et al., 2020), and SynFlow (Tanaka et al., 2020) are selected, which locates sparse subnetworks at random initialization via certain salience criterion; (5) *dynamic sparse training (DST)*. We choose the top-performing algorithm, RigL (Evci et al., 2020; Liu et al., 2021b), which starts from a random sparse network and encourages the connectivity to evolve dynamically based on a grow-and-prune strategy. All results and analyses about RigL are referred to Appendix A2.2.

**Implementation details.** Experiments are conducted on CIFAR-10 (C10), CIFAR-100 (C100) (Krizhevsky & Hinton, 2009), and ImageNet (IMG) (Deng et al., 2009). For a fair comparison, we follow the standard implementations and hyperparameters in (Renda et al., 2020) for OMP, LTH, RP, and PI experiments, as shown in Table 1. All RigL experiments follow the recent SOTA training configurations (Liu et al., 2021b). More details can be found in the Appendix A1.

## 4 What is Lost or Gained after Pruning?

In this section, we comprehensively investigate the full-dimension performance of sparse neural networks from LTH and other pruning algorithms, including (*i*) generalization to distribution shifts, (*ii*) uncertainty and reliability, (*iii*) interpretability, and (*iv*) geometry of loss landscapes. If a sparse subnetwork $m \odot \theta$ can be trained from the random initialization and match the dense model results in aspects ($i - iv$), it is a `LTH-PASS`. The detailed sparsity levels of `LTH-PASS` in our scenarios are provided in Appendix A2.4.

*In summary, `LTH-PASS` broadly exists for diverse network architectures and datasets at $20 \sim 99.53\%$ sparsity.*

### 4.1 Generalization to Distribution Shifts

The generalization ability of (sparse) neural networks is often considered equal to their training-test accuracy gap, while the test sets are i.i.d. selected from the same underlying distribution as the training set. While sparse neural networks can often achieve unimpaired test set accuracy, we broaden the scope of generalization ability

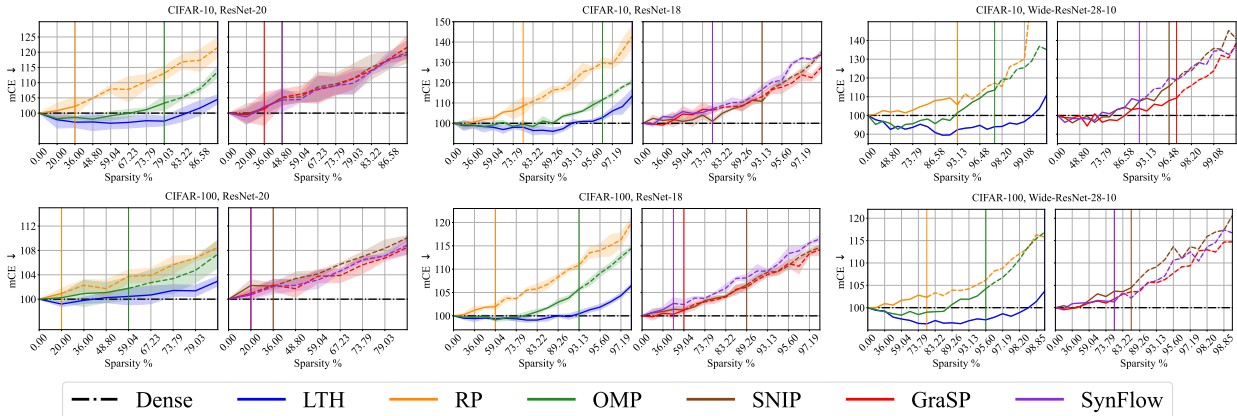

Figure 3: **Natural corruption robustness**, i.e. mCE ($\downarrow$), of diverse subnetworks with a range of sparsity from 0.00% to 99.53% on CIFAR-10-C and CIFAR-100-C datasets.

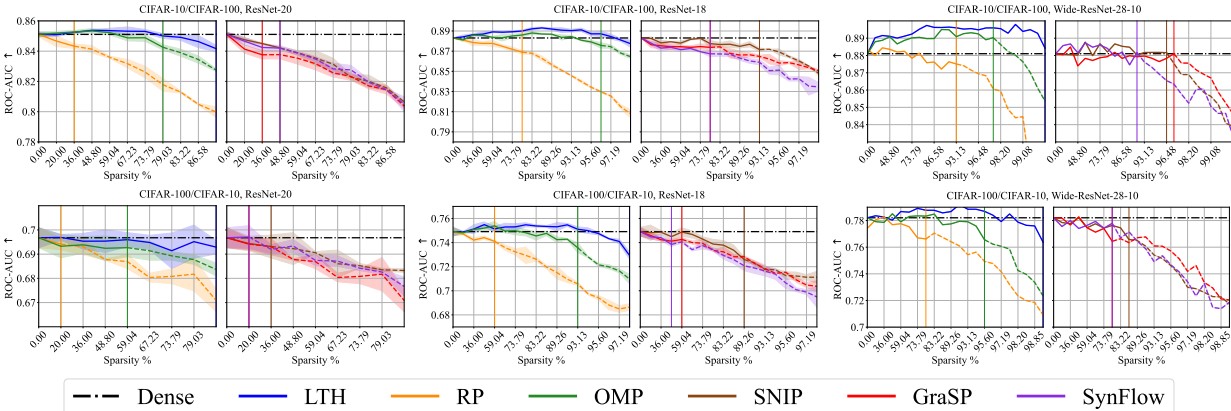

Figure 4: **Out-of-distribution** (OoD) detection performance, i.e., ROC-AUC ($\uparrow$), of diverse subnetworks with a range of sparsity from 0.00% to 99.53% on CIFAR datasets. Titles are formed by in-distribution dataset / out-of-distribution dataset, architecture.

by considering manipulated or shifted data distributions. Specifically, we will examine their generalization to *natural corruptions*, *adversarial perturbations* and *out-of-distribution (OoD) data* performance. The main takeaways are summarized below.

> **Takeaways: ❶** With appropriate sparsity levels, e.g., $59.04\% \sim 99.53\%$, subnetworks from LTH enjoy better generalization than its dense models, on (shifted) data distributions. **❷** Regardless of pruning methods, sparse neural networks are relatively more brittle to natural corruptions, compared to the other data distribution shifts.

Specifically, we quantify the generalization ability of network pruning in four main aspects:

● (Clean) generalization gap: $\frac{1}{|\mathcal{D}_{\text{test}}|} \sum_{(x,y)\in\mathcal{D}_{\text{val}}} \delta(f(x;\theta)=y) - \frac{1}{|\mathcal{D}_{\text{train}}|} \sum_{(x,y)\in\mathcal{D}_{\text{train}}} \delta(f(x;\theta)=y)$, where $f(x;\theta)$ is the model's output and $\delta(\cdot)$ is the indicator function. $\mathcal{D}_{\text{train}}$, $\mathcal{D}_{\text{test}}$, $x$ and $y$ denotes the training data, testing data, input sample, and its corresponding label. Empirically, for well-trained models (i.e., $\sim$ zero training error), *the test set accuracy* is adopted to represent the generalization ability on original test sets, which is the conventional metric to evaluate the quality of sparse neural networks.

● Natural corruption robustness: mCE $= \frac{1}{|\mathcal{C}|} \sum_{c\in\mathcal{C}} [\sum_{s=1}^{5} \mathcal{E}_{s,c}^{m\odot\theta}) / (\sum_{s=1}^{5} \mathcal{E}_{s,c}^{\theta})]$. Following the standard setup in Hendrycks & Dietterich (2019), we use the mean corruption error (mCE) to indicate model robustness to different natural corruptions, where $\mathcal{E}_{s,c}^{m\odot\theta}$ and $\mathcal{E}_{s,c}^{\theta}$ are the top-1 error of dense model $\theta$ and its sparse subnetwork $m \odot \theta$, respectively. $\mathcal{C}$ is the set of corruptions such as noise, blur, weather, digital process, and each corruption type $c \in \mathcal{C}$ has five corruption severity levels (i.e., $1 \le s \le 5$). Note that all corrupted samples

are never shown in the training stage. CIFAR-10/100-C and ImageNet-C (Hendrycks & Dietterich, 2019) are adopted in our experiments. More details are included in Appendix A1.

• Adversarial robustness: testing accuracy on adversarial perturbed images, i.e., robust accuracy. We choose the classical adversarial attack, i.e., Fast Gradient Sign Method (FSGM) (Goodfellow et al., 2014), to generate adversarial samples as $x + \epsilon \times \text{sgn}(\nabla_x \mathcal{L}(f(x; \theta), y))$, where $\mathcal{L}$ is the empirical loss and $\epsilon$ (in our case, $\epsilon = \frac{8}{255}$) is the predefined magnitude of perturbations.

• Out-of-distribution (OoD) performance: ROC-AUC[1] as the standard metric is utilized to gauge obtained subnetworks. Since deep neural networks suffer from overconfident predictions on out-of-distribution data (Nguyen et al., 2015; Hendrycks & Gimpel, 2016), it is valuable to investigate whether this issue will be amplified or diminished by introduced model sparsity. Following Hendrycks & Gimpel (2016); Hendrycks et al. (2018); Hein et al. (2019); Augustin et al. (2020), for CIFAR-10 experiments, CIFAR-100 (Krizhevsky & Hinton, 2009) is regarded as the OoD dataset; for CIFAR-100 experiments, CIFAR-10 is selected as the OoD dataset; for ImageNet experiments, ImageNet-O (Hendrycks et al., 2019) is the OoD dataset.

**Experimental observations.** We present the results of **test accuracy**, **natural corruption robustness**, **OoD performance**, and **adversarial robustness** in Figure 2, 3, 4, and A14, respectively. Additional results of (IMG, R50), (C10, VGG-11), and (C100, VGG-11) can be found in 10, A12, and A13, respectively. Several consistent findings can be drawn:

① *Superior sparse models?* Winning tickets broadly exist with unimpaired generalization on multiple (shifted) data distributions. Specifically, the matched or even outperformed performance can be achieved by winning tickets at sparsity $\{83.22 \sim 86.58\%, 93.13\% \sim 98.20\%, 94.50\% \sim 99.53\%, 67.23\% \sim 83.22\%, 93.13\% \sim 97.75\%, 98.20\% \sim 99.08\%, 59.04\% \sim 89.26\%\}$[2] on the original, corrupted, adversarial perturbed, out-of-distribution data. We see the extreme sparsity of winning tickets alters substantially given different evaluation metrics like natural corruption robustness, which suggests the limitation of considering the clean test set accuracy as the only quality measurement for pruned subnetworks.

② *Sensitivity metrics?* In general, natural corruption robustness (mCE) is the most sensitive measure to pruning since found winning tickets via mCE have smaller extreme sparsity. It suggests excessively pruned models are relatively more fragile to natural corruptions, such as blur, noise, fog, etc., which coincides with the findings in Hooker et al. (2020b).

③ *Data or model dependent?* Regarding to investigated generalization on various (shifted) data distributions, on the same dataset, more overparameterized models (e.g., WR28-10 v.s. R20s) are more amenable to be sparsified; with the same network, dataset contains more classes (e.g., C100 v.s. C10) is more intractable for pruning. Similar observations also presented in Morcos et al. (2019a).

④ *Superior pruning methods?* With low sparsity levels (e.g., $\leq 48.80\%$), OMP appears comparable generalization ability to LTH, and all PI algorithms perform no better than random pruning. At higher sparsity levels, although PI especially GraSP shows moderate advantages of generalization on the original test set and out-of-distribution data, subnetworks from PI are similarly vulnerable to corrupted or perturbed samples as randomly pruned models.

## 4.2 Calibration and Reliability

*Confidence calibration* uncovers the prediction uncertainties and the model reliability (Guo et al., 2017; Quiñonero-Candela et al., 2006; DeGroot & Fienberg, 1983; Venkatesh et al., 2020). It advocates the classification models must not only be accurate but also should reveal the true correctness likelihood (i.e., when the predictions are likely to be incorrect) (Guo et al., 2017), especially in safety/security-critical scenarios like self-driving vehicles (Bojarski et al., 2016) and automated health care (Jiang et al., 2012). In this section, we investigate whether pruning hurts or benefits confidence calibration. The main takeaways are summarized below.

---

[1]ROC-AUC stands for the area under the receiver operating characteristic (ROC) curve, in which we adopt the prediction confidence as the threshold.

[2]Results are produced by the configurations of {(C10,R20s), (C10,R18), (C10,WR28-10), (C100, R20s), (C100,R18), (C100,WR28-10), (IMG,R50)}. Such narrative style is adopted hereinafter.

> **Takeaways:** ❶ Sparse networks from LTH with $48.80\% \sim 99.53\%$ sparsity, are capable of maintaining or enhancing both generalization and uncertainty performance, compared to dense models. ❷ In general, uncertainty measures are more sensitive to pruning than generalization metrics.

Several classical and representative evaluation metrics (Guo et al., 2017; Venkatesh et al., 2020) are used in our experiments:

• Expected calibration error (ECE): ECE $= \sum_{m=1}^{M} \frac{|B_m|}{n} |\text{acc}(B_m) - \text{conf}(B_m)|$, where $n$ and $|B_m|$ is the number of total samples and samples in the bin $B_m$, respectively. ECE (Pakdaman Naeini et al., 2015) is a widely adopted metric to approximate the difference in expectation between confidence and accuracy (i.e., miscalibration). Specifically, it partitions predictions into $M$ equally-spaced bins, and then calculate a weighted average of the accuracy/confidence discrepancy in each of these bins.

• Static calibration error (SCE): SCE $= \frac{1}{nC} \sum_{c=1}^{C} \sum_{m=1}^{M} |\sum_{i \in B_m} \mathbf{1}(y_i = c) - \text{conf}(B_m)|$, where $C$ is the total number of classes and $y_i$ denotes the label of sample $i$. SCE bins the predictions separately for each class probability. Unlike ECE that only considers the highest probability, SCE (Gweon & Yu, 2019) treats all probabilities in a multi-class regime equally. We adopt it for ImageNet experiments with $1,000$ classes.

• Negative log likelihood (NLL): NLL $= -\sum_{(x,y) \in \mathcal{D}_{\text{val}}} \log(\hat{p}(y|x))$ as another standard measure of the calibration quality (Hastie et al., 2001; LeCun et al., 2015), is minimized in expectation if and only if the prediction distribution $\hat{p}(Y|X)$ recovers the ground truth conditional distribution $p(Y|X)$.

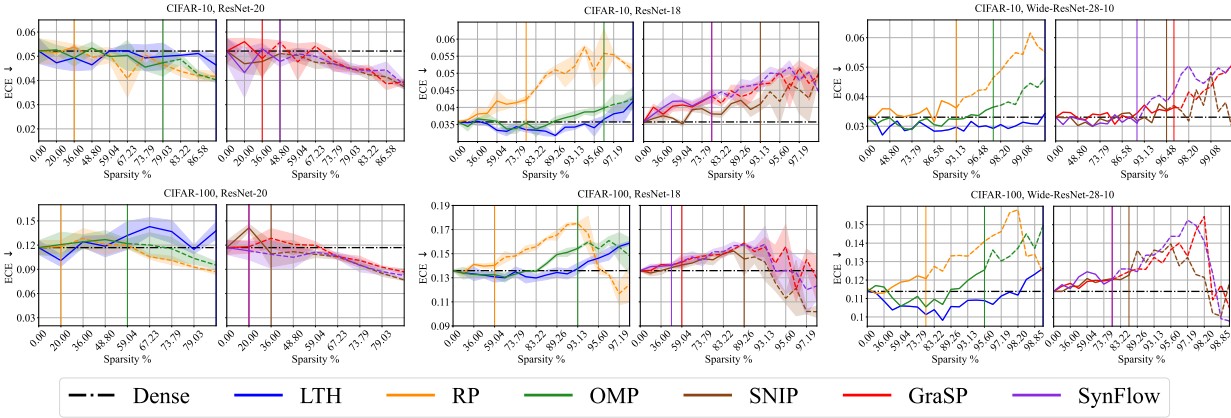

Figure 5: **Expected calibration error** (ECE ↓) of diverse subnetworks with a range of sparsity from 0.00% to 99.53% on CIFAR datasets. More results can be found in Figure A12 and A13.

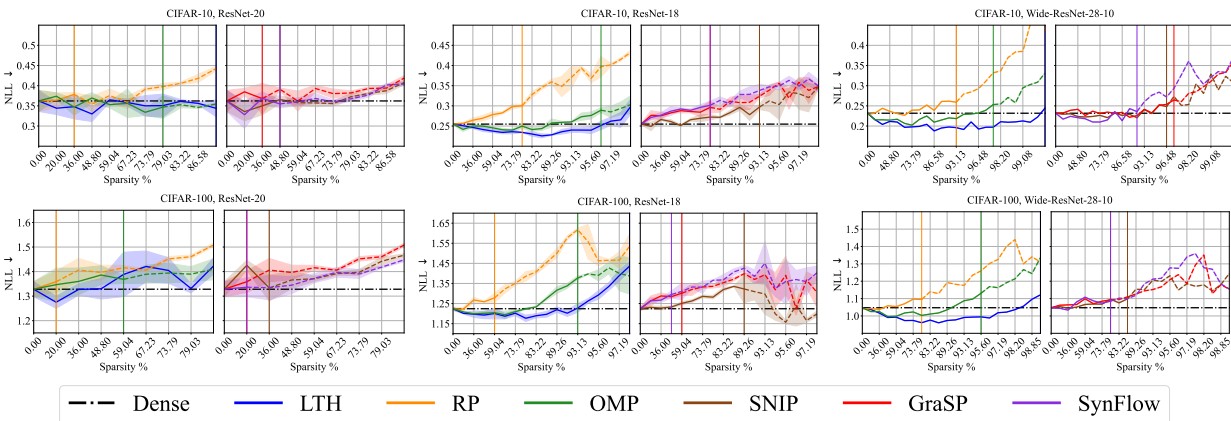

Figure 6: **Negative log likelihood** (NLL ↓) of diverse subnetworks with a range of sparsity from 0.00% to 99.53% on CIFAR datasets, which is normalized by dense networks.

**Experimental observations.** Main results of **expected calibration error** and **negative log likelihood** are presented in Figure 5 and 6. More results of (IMG, R50), (C10, VGG-11), and (C100, VGG-11) are included in Figure 10, A12, and A13 respectively. We observe that:

① *Superior sparse models?* Winning tickets in terms of uncertainty metrics can be found at sparsity $\{89.26\%, 96.48 \sim 97.75\%, 99.53\%, 48.80\%, 93.13\%, 98.20\%, 48.80\% \sim 73.09\%\}$.

② *Sensitivity metrics?* Generally, ECE and NLL metrics show a similar sensitivity to pruning. While uncertainty measures are more sensitive than generalization measures, it is within expectation since investigated pruning algorithms are mainly designed to avoid generalization drops.

③ *Superior pruning methods?* At high sparsity ratios (e.g., $\geq 67.23\%$), RP and PI have outperformed ECE than LTH in cases (C10/C100,R20s), yet at the price of high degraded generalization ability. Similar things happened in exorbitantly sparsified models in cases (C100,R18/WR28-10).

### 4.3 Interpretability

In this section, we investigate whether and which sparse neural networks are able to maintain the interpretability, compared to their dense counterpart. We assess these subnetworks from both macro and micro views. The former quantitatively evaluates the explainability from the functional representation perspective, with the most commonly used metrics, i.e., *fidelity* (Plumb et al., 2020; 2018; Ribeiro et al., 2016) and *stability* (Plumb et al., 2020; Alvarez-Melis & Jaakkola, 2018; Ghorbani et al., 2019). The latter performs the NetDissect procedure (Bau et al., 2017) for explaining neurons' behavior by identifying compositional logical concepts (Mu & Andreas, 2020). The main takeaways lie below.

---

**Takeaways:** ❶ Although LTH has superior generalization and uncertainty performance than other pruning methods, it is much more difficult to be interpreted by linear explainers. ❷ When we dissect sparse neural networks, LTH shows a significantly enhanced interpretability in terms of neuron behaviors, even compared to its dense counterpart.

---

• Fidelity and stability: $\mathcal{F} = \mathbb{E}_{x \in \mathcal{D}_{\text{val}}}[\mathbb{E}_{x' \sim \mathcal{N}_x}[(g(x') - f(x'))^2]], \mathcal{S} = \mathbb{E}_{x \in \mathcal{D}_{\text{val}}}[\mathbb{E}_{x' \sim \mathcal{N}_x}[\|e(x, f) - e(x', f)\|_2^2]]$. *Fidelity* $\mathcal{F}$ and *stability* $\mathcal{S}$ focus on local explanations for semantic features, which attempts to predict how the model's output would change if the input samples were perturbed. Following the classical routines in LIME (Ribeiro et al., 2016) to compute the metrics, we first perturb each input images $x$ and build its neighborhood set $\mathcal{N}_x$ with a size of $1,000$ samples. Then, we generate a class of interpretable functions $\mathcal{G} := \{g_x \in \mathcal{G} | x \in \mathcal{D}_{\text{val}}\}$, where $g_x$ is a linear function obtained form a regression to the corresponding model's output on $\mathcal{N}_x$. In the above formulation, $f(\cdot)$ denotes the target model we want to interpret. $e(x, f), e(x', f)$ are the learned weights of linear models $g_x$ and $\tilde{g}_x$. Both $g_x$ and $\tilde{g}_x$ are trained on $\mathcal{N}_x$, while each training sample $\hat{x} \in \mathcal{N}_x$ is weighted by the Hamming distance of $(\hat{x}, x)$ and $(\hat{x}, x')$ (Ribeiro et al., 2016), respectively. All our experiments follows the same implementation in (Plumb et al., 2020). Intuitively, $\mathcal{F}$ quantifies how accurately the explainer $g_x$ models the target network $f$ in a neighborhood $\mathcal{N}_x$; $\mathcal{S}$ measures the degree to which the explanation changes across points in $\mathcal{N}_x$.

• Composition neurons. Following (Mu & Andreas, 2020), we consider each individual neuron $f_n(x) \in \mathbb{R}$ of the model's output $f(x) \in \mathbb{R}^{d_2}$, and its activation on concrete input images. A good explanation of neuron $f_n$ is a description (e.g., category or property) which locates the same inputs for which $f_n$ activates. Specifically, we search the most appropriate *compositional concepts* $\mathcal{K}$ with the largest IoU (i.e., the intersection over union) for neuron $f_n$, i.e., $\text{IoU}(n, \mathcal{K}) = [\sum_x \mathbb{1}(M_n(x) \wedge \mathcal{K}(x))]/[\sum_x \mathbb{1}(M_n(x) \vee \mathcal{K}(x))]$, where $M_n(x)$ is a binary mask generated by thresholding the continuous neurons activation of $f_n(x)$. $\mathcal{K}$ consists of several pre-defined atomic concepts $\mathcal{K}_i$ ($1 \leq i \leq t$, $t$ is 3 in our experiments) from the ADE20k (Zhou et al., 2017) and Broden (Bau et al., 2017) datasets. Each atomic concept is an image segmentation mask, and can be combined via disjunction (OR), conjunction (AND), and negation (NOT) operations, as shown in Figure 7. All procedures of our experiments follow the default configuration in (Bau et al., 2017; Mu & Andreas, 2020), and models for NetDissect are (sparse) R50 trained on ImageNet.

**Experimental observations.** According to main results in Figure 8, 9, and 7, we summarize important observations below. Additional experimental results are presented in Appendix A2. Figure A12 and A13

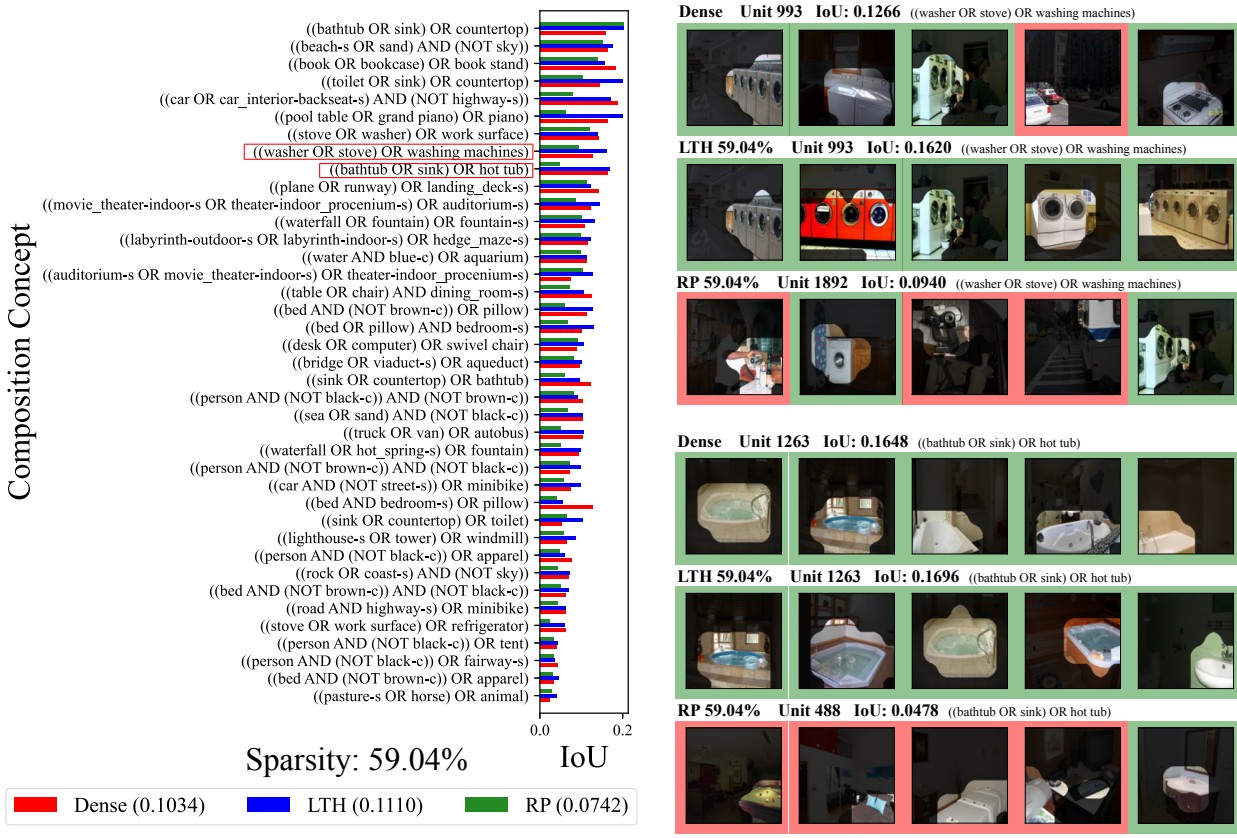

Figure 7: Results of **composition neurons** with R50 on ImageNet. (*Left*) Maximum IoUs (↑) of the intersection compositional concepts (top 70) by dissecting dense (ave. IoU: 0.1034), winning tickets (ave. IoU: 0.1110), and randomly pruned networks (ave. IoU: 0.0742) at the 59.04% sparsity. (*Right*) Neuron explanations for two of top IoU concepts in the left figure's red box. For each concept and neuron, top-5 IoU samples are presented. Green border indicates objects are **coincided** with the concept. Red border means they are **unrelated**.

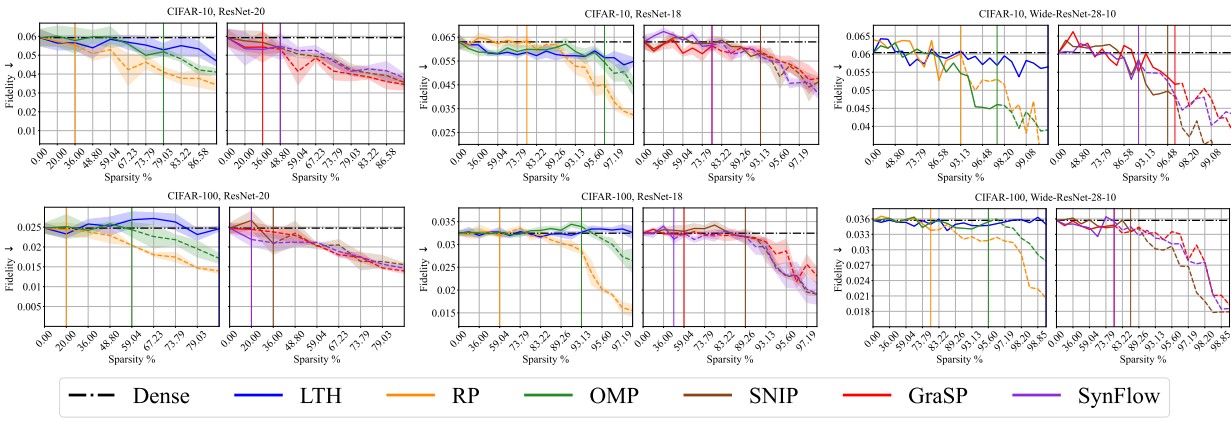

Figure 8: **Fidelity** (↓) of diverse subnetworks with a range of sparsity from 0.00% to 99.53% on CIFAR.

collect the evaluation on (C10, VGG-11) and (C100, VGG-11), respectively. Figure A18 and A19 display the extra composition neuron results of subnetworks at 67.23% sparsity. Figure A17 reports the IoU distribution of all 2048 neurons of diverse sparse neural networks.

① *Linear interpretability.* On CIFAR, we can locate winning tickets with a range of sparsity from 20% to 99.53%. Meanwhile, although with an inferior generalization, RP, OMP, and PI algorithms have consistently

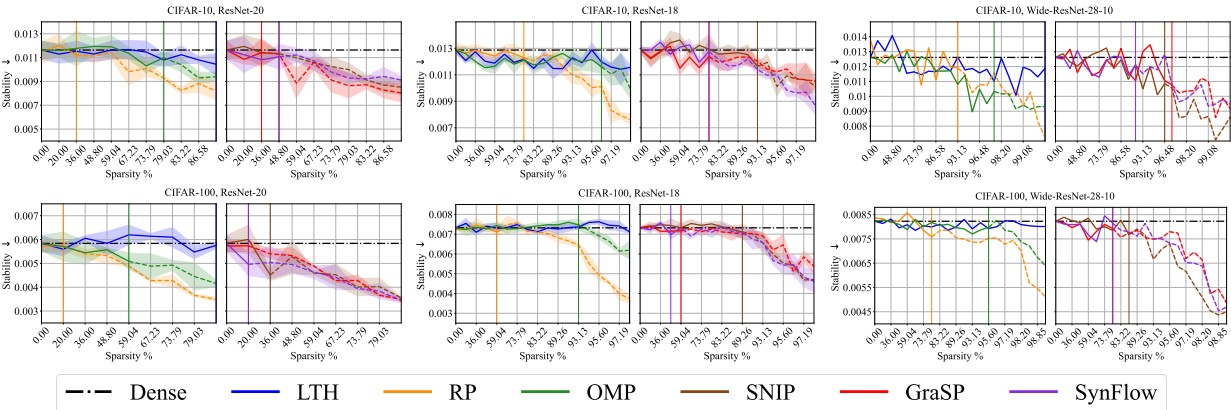

Figure 9: **Stability** (↓) of diverse subnetworks with a range of sparsity from 0.00% to 99.53% on CIFAR.

better linear interpretability. It suggests that the critical sparse topology mined by LTH potentially offers a more non-linear functional representation.

➋ *Neuron behaviors.* Based on NetDissect (Bau et al., 2017; Mu & Andreas, 2020) results in Figure 7 and A17, we find winning tickets (LTH) at 59.04% sparsity to achieve competitive generalization and neuron interpretability, compared to the unpruned model. Coherently with the observations in (Mu & Andreas, 2020), when a neuron is active, the more generalizable the sparse network is, the more interpretable that neuron is (with a large IoU). Qualitative visual results in Figure 7 also imply the winning tickets from LTH succeed in finding more interpretable true positives than the corresponding dense model and randomly pruned networks.

## 4.4 Geometry of Loss Landscapes

The geometry of loss surfaces (e.g., *flatness*) reflects the learned functional approximation of derived subnetworks, which provides various insights to assess the sparse model's generalization ability (Evci et al., 2022) and understand its behaviors such as transferability (Liu et al., 2019a). Some other works (Hochreiter & Schmidhuber, 1997; Keskar et al., 2017; Jiang et al., 2019) show that the loss landscapes of well-generalizing models are relatively "flat" respect to model weights. Similarly, (Wu et al., 2020; Moosavi-Dezfooli et al., 2019; Chen et al., 2021d) claim that a flatter adversarial loss landscape with respect to model inputs enhances the robustness generalization. The weight/input flatness are defined as the Hessian of objective function respect to the weight/input samples, respectively. Our main takeaways can be summarized as follows. More details are referred to Appendix A2.2.

> **Takeaways:** ➊ Winning tickets (LTH) with 20.00% ∼ 99.53% sparsity exist, achieving unscathed generalization ability, uncertainty, interpretability, and the loss landscape geometry. ➋ Besides pruning methods, network backbones and dataset scale also play non-negligible roles in the learned loss geometry of sparse neural networks.

## 4.5 Extra Experimental Investigations

**ImageNet results.** Figure 10 presents the evaluation results of ResNet-50 on ImageNet with a range of sparsity from 0.00% to 93.13%, in which the PR-AUC metric stands for the area under the precision and recall (PR) curve. We can observe that subnetworks from LTH are capable of maintaining or enhancing the performance of all the metrics with appropriate sparsity levels (which are the `LTH-PASS`), e.g., 48.80% ∼ 93.13%, in which LTH consistently outperforms both PI methods and RP.

**Extra network backbone.** In this section, we evaluate an additional network architecture, VGG-11 (Simonyan & Zisserman, 2014), beyond ResNets. Results on CIFAR-10 and CIFAR-100 are collected in Figure A12 and A13 respectively. We observe that winning lottery tickets exist at (98.20%, 96.48%, 98.20%,

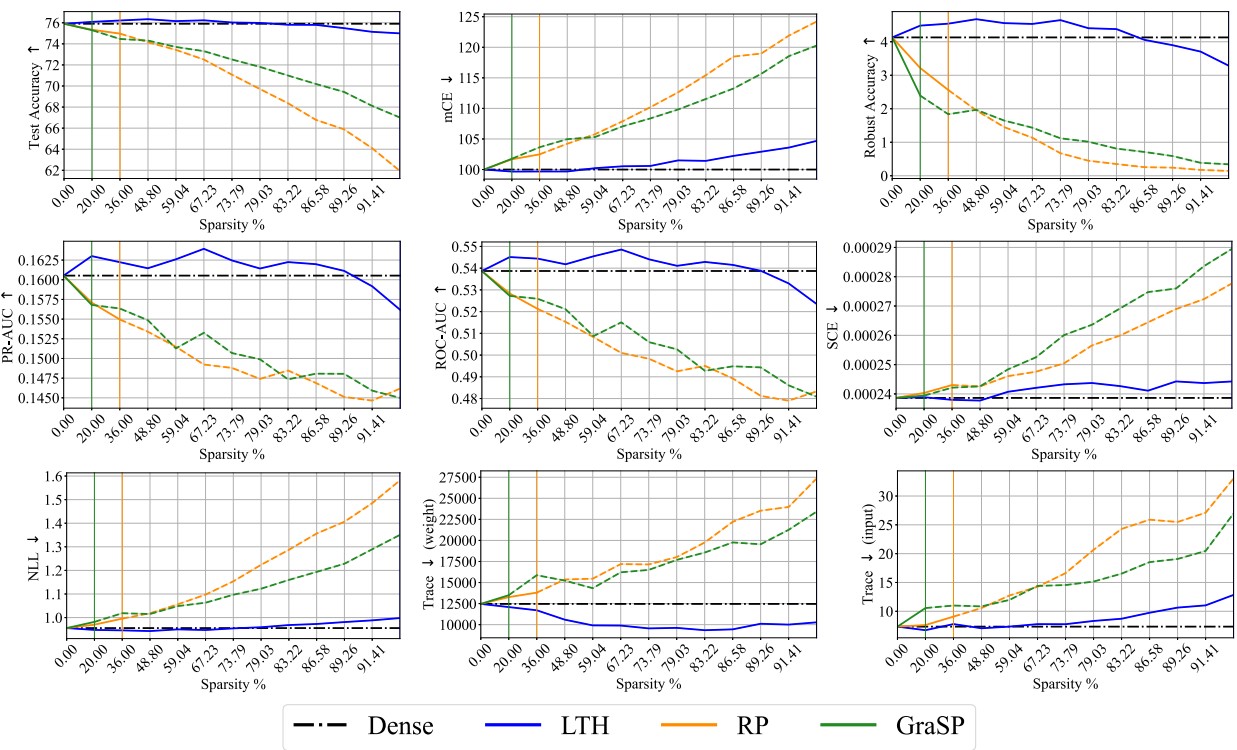

Figure 10: Evaluation results of ResNet-50 on ImageNet, including **Generalization ability** (Test Accuracy, Natural Corruption Robustness, Adversarial Robustness, Out-of-distribution Detection Performance (ROC-AUC)), **Uncertainty** (ECE, NLL), **Interpretability** (Fidelity, Stability) and **Geometric of Loss Surfaces** (Trace of Hessian Matrix).

98.20%) and (96.48%, 89.26%, 96.48%, 95.60%) sparsity levels in terms of generalization ability (test accuracy, natural corruption robustness, adversarial robustness, out-of-distribution detection performance), (98.20%, 98.20%) and (86.58%, 94.50%) sparsity levels in terms of uncertainty (ECE, NLL), (98.20%, 98.20%) and (91.41%, 96.48%) sparsity levels in terms of interpretability (fidelity, stability), (98.20%) and (93.13%) sparsity levels in terms of loss surfaces' geometric, for (C10, VGG-11) and (C100, VGG-11) respectively. Therefore, `LTH-PASS` is identified in these two cases with the sparsity of 96.48% and 86.58%.

**Comparison with a smaller dense network.** As demonstrated in Figure 11, we observe the `LTH-PASS` at the 79% sparsity outperforms the small-dense baseline with similar parameter counts, by significant performance margins along all four evaluation dimensions. It suggests the sparsity functions as a comprehensive regularization and influences diverse aspects of neural networks, far beyond a simple reduction of network capacity (i.e., parameter counts).

**Comparison to dynamic sparsity.** From Figure 11, we observe that LTH and RigL are capable of maintaining the generalization ability of dense networks at a sparsity level of 21%. As for the uncertainty measurement, LTH and RigL show better performance than dense networks. On CIFAR-10 and CIFAR-100, all sparse networks have better linear interpretability than dense networks. And with the scope of Hessian traces for weight flatness, the subnetworks located by LTH are winning tickets on CIFAR-10 and CIFAR-100, while RigL fails to locate the flat local minima. Overall, identifying sparse neural networks from dynamic sparse training (e.g., RigL) is a great option for preserving generalization ability, interpretability, and uncertainty, but not for maintaining flat geometric of learned loss surfaces.

## 5 Conclusion and Discussion of Broad Impact

In this paper, we perform an exhaustive screening test on the performance of various sparse neural networks, along diverse dimensions far beyond test-set accuracy. We believe that our compelling empirical results

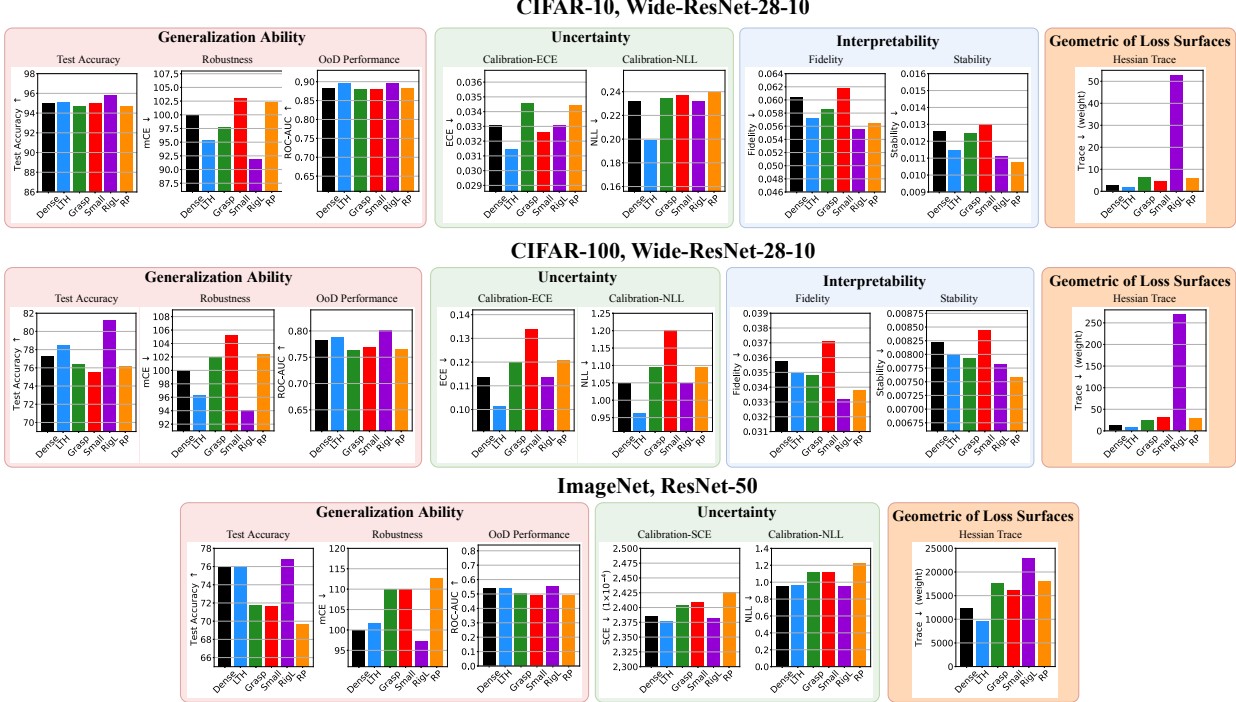

Figure 11: Evaluation results of sparse neural networks at sparsity level of 79% on CIFAR-10, CIFAR-100 and ImageNet. **Dense** is the unpruned full model; LTH, Grasp, RigL and RP denote the sparse networks identified by the LTH, Grasp, dynamic sparse training and random pruning, respectively. Small represents a smaller dense network of similar parameter counts to sparse networks.

offer many in-depth insights of understanding network pruning, and endorse the wider adoption of (properly chosen) sparse neural networks in place of dense ones. Our future works will extend the similar screening to other model compression methods such as quantization.

We do not think this scientific research places a substantial risk of societal harm. The potential societal impact is that, with the assistance of our comprehensive assessment, it may be possible to establish accurate, robust, reliable, interpretable sparse networks with reduced energy and financial costs.

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
