# OpenReview forum: "Can You Win Everything with A Lottery Ticket?"
_TMLR — Accepted by TMLR_

### Review · Reviewer_FNDL · 2022-07-23

**Summary Of Contributions:**

This empirical paper shows that for standard image classification tasks, one could find a rather sparse network that improves eight different categories of model performance: 1. test accuracy; 2. robustness; 3. OoD performance; 4. Flatness; 5. Composition Neurons; 6. Fidelity; 7. Stability; 8. Calibration

**Broader Impact Concerns:**

Fine.

**Requested Changes:**


1. Clarify the two weaknesses I raised, and change the title to a proper one if my criticism applies (otherwise, explain why my criticism is incorrect)
- I cannot recommend acceptance for a work whose title is either overclaiming or misleading.

2. Definition of "sparsity" needs to be explicitly given (and at a highly visible place), given that it is the x-axis of every plot

**Strengths And Weaknesses:**

Strength: the experiment is though and convincing. Initially, I felt that the specific reference to the "lottery ticket hypothesis" is a bit too strong, but the experiment suggests that most of the demonstrated advantage does appear in the strong sparsity range (with above 80% sparsity), so I think the reference to lottery ticket is fine. However, I cannot decide before the authors answer to my following questions


Weakness: there are a few problems that I think need clarification:
1. for a fixed method, is it the case that there exists a single sparse model that achieves LTH-PASS simultaneously? This point is not clear. From the experiments, it seems like this is not the case -- for Fig. 2 the best test ACC is achieved at roughly 50% sparsity, whereas in Fig. 10, the improved flatness only appears at roughly 75% sparsity.

2. If the answer to question 1 is no, I feel that the title is misleading and needs to be changed. Different models and different levels of sparsity seem to be required to achieve the eight different categories. Therefore, it does not take ONE lottery ticket to "win everything" but takes EIGHT different lottery tickets to "win everything." I feel that there is a severe overclaim in the title (at least, I think the title is sufficiently misleading).

---

> ### Author Response · Authors · 2022-07-27
> **Response to Reviewer FNDL [Cons 1]**
>
> Many thanks for acknowledging our experiments as though and convincing. We point-wisely address all your concerns as below:
>
> **[Cons 1. Single model win all?]** **It is ONE lottery ticket to “win everything” rather than EIGHT different lottery tickets**. For a fixed method, network backbone, and dataset, there exists a single sparse model that achieves LTH-PASS **simultaneously**.
> Specifically, we perform certain pruning algorithms to obtain a series of sparse models, and then evaluate each of them under all metrics. If there is a single sparse model that reaches the matched performance of its dense counterpart across all the evaluation metrics, we call it LTH-PASS.
>
> From our results, we observe:
>
> - In **(ResNet-20, CIFAR-10)**, the sparse subnetwork from LTH has undamaged performance compared to its dense counterpart, with sparsity levels less than {89.26%, 83.22%, 86.58%, 86.58%, 89.26%, 89.26%, 89.26%, 89.26%, 86.58%, 83.22%} in terms of {clean test accuracy, natural corruption robustness, adversarial robustness, ROC-AUC, ECE, NLL, fidelity, stability, Hessian trace-weight, Hessian trace-input} respectively. Therefore, the LTH-PASS (a single sparse model) exists at sparsity levels less than 83.22% and wins “everything”.
>
> - In **(ResNet-18, CIFAR-10)**, the sparse subnetwork from LTH has undamaged performance compared to its dense counterpart, with sparsity levels less than {98.20%, 93.13%, 98.20%, 97.75%, 96.48%, 97.75%, 98.20%, 98.20%, 97.75%, 96.48%} in terms of {clean test accuracy, natural corruption robustness, adversarial robustness, ROC-AUC, ECE, NLL, fidelity, stability, Hessian trace-weight, Hessian trace-input} respectively. Therefore, the LTH-PASS (a single sparse model) exists at sparsity levels less than 93.13% and wins “everything”.
>
> - In **(Wide-ResNet-28-10, CIFAR-10)**, the sparse subnetwork from LTH has undamaged performance compared to its dense counterpart, with sparsity levels less than {99.53%, 99.26%, 94.50%, 99.53%, 99.53%, 99.53%, 99.53%, 99.53%, 99.53%, 99.41%} in terms of {clean test accuracy, natural corruption robustness, adversarial robustness, ROC-AUC, ECE, NLL, fidelity, stability, Hessian trace-weight, Hessian trace-input} respectively. Therefore, the LTH-PASS (a single sparse model) exists at sparsity levels less than 94.50% and wins “everything”.
>
> - In **(VGG-11, CIFAR-10)**, the sparse subnetwork from LTH has undamaged performance compared to its dense counterpart, with sparsity levels less than {98.20%, 96.48%, 98.20%, 98.20%, 98.20%, 98.20%, 98.20%, 98.20%, 98.20%} in terms of {clean test accuracy, natural corruption robustness, adversarial robustness, ROC-AUC, ECE, NLL, fidelity, stability, Hessian trace-weight} respectively. Therefore, the LTH-PASS (a single sparse model) exists at sparsity levels less than 96.48% and wins “everything”.
>
> - In **(ResNet-20, CIFAR-100)**, the sparse subnetwork from LTH has undamaged performance compared to its dense counterpart, with sparsity levels less than {83.22%, 67.23%, 83.22%, 83.22%, 48.80%, 48.80%, 20.00%, 20.00%, 79.03%, 20.00%} in terms of {clean test accuracy, natural corruption robustness, adversarial robustness, ROC-AUC, ECE, NLL, fidelity, stability, Hessian trace-weight, Hessian trace-input} respectively. Therefore, the LTH-PASS (a single sparse model) exists at sparsity levels less than 20.00% and wins “everything”.
>
> - In **(ResNet-18, CIFAR-100)**, the sparse subnetwork from LTH has undamaged performance compared to its dense counterpart, with sparsity levels less than {97.75%, 93.13%, 97.75%, 95.60%, 93.13%, 93.13%, 93.13%, 93.13%, 97.19%, 97.19%} in terms of {clean test accuracy, natural corruption robustness, adversarial robustness, ROC-AUC, ECE, NLL, fidelity, stability, Hessian trace-weight, Hessian trace-input} respectively. Therefore, the LTH-PASS (a single sparse model) exists at sparsity levels less than 93.13% and wins “everything”.
>
> - In **(Wide-ResNet-28-10, CIFAR-100)**, the sparse subnetwork from LTH has undamaged performance compared to its dense counterpart, with sparsity levels less than {99.08%, 98.56%, 99.08%, 98.20%, 98.20%, 98.20%, 99.08%, 99.08%, 98.85%, 98.85%} in terms of {clean test accuracy, natural corruption robustness, adversarial robustness, ROC-AUC, ECE, NLL, fidelity, stability, Hessian trace-weight, Hessian trace-input} respectively. Therefore, the LTH-PASS (a single sparse model) exists at sparsity levels less than 98.20% and wins “everything”.

---

> ### Author Response · Authors · 2022-07-27
> **Response to Reviewer FNDL [Cons 1-2]**
>
> **[Cons 1. Continued]**
>
> - In **(VGG-11, CIFAR-100)**, the sparse subnetwork from LTH has undamaged performance compared to its dense counterpart, with sparsity levels less than {96.48%, 89.26%, 96.48%, 95.60%, 86.58%, 94.50%, 91.41%, 96.48%, 93.13%} in terms of {clean test accuracy, natural corruption robustness, adversarial robustness, ROC-AUC, ECE, NLL, fidelity, stability, Hessian trace-weight} respectively. Therefore, the LTH-PASS (a single sparse model) exists at sparsity levels less than 86.58% and wins “everything”.
>
> - In **(ResNet-50, ImageNet)**, the sparse subnetwork from LTH has undamaged performance compared to its dense counterpart, with sparsity levels less than {93.13%, 59.04%, 86.58%, 89.26%, 48.80%, 79.03%, 59.04%, 93.13%, 59.04%} in terms of {clean test accuracy, natural corruption robustness, adversarial robustness, ROC-AUC, ECE, NLL, composition neurons, Hessian trace-weight, Hessian trace-input} respectively. Therefore, the LTH-PASS (a single sparse model) exists at sparsity levels less than 48.80% and wins “everything”.
>
> “whereas in Fig. 10, the improved flatness only appears at roughly 75% sparsity.” We politely point out that it is a misunderstanding. In Figure 10, a smaller trace (↓) indicates a better flatness. Therefore, all subnetworks have the improved/undamaged flatness at 20% sparsity, and most of them still win flatness at 50% sparsity.  The above clarifications will be included in our final version.
>
> **[Cons 2. Definition of sparsity.]** Great suggestion. We will add the definition of “sparsity” explicitly to Section 3 of Preliminary as follows:
>
> “Sparsity of a compressed neural network is defined as the ratio of removed parameters to the total parameters, e.g., sparsity = 1-||m||_0 / d in our case.” ||m||_0 is the number of non-zero elements and d is the total number of parameters.

---

### Review · Reviewer_jy8z · 2022-07-25

**Summary Of Contributions:**

This work extensively studies sparse neural networks beyond the test-set accuracies on ResNet architectures train on Cifar 10/100 and Imagenet. Studied dimensions are intriguingly many: (1) natural-corruptions/OOD generalization/adverserial robustness (2) calibration error, (3) interpretability and (4) flatness of the solution (See Figure-1). Overall these dimensions seem to correlate highly with test accuracy (especially on Imagenet). But more importantly this work confirms that sparse networks are better on all dimensions than the dense networks at the same parameter count, which I believe is useful for the community

**Broader Impact Concerns:**

No concern.

**Requested Changes:**

# Major
- (critical) I'm not sure what would be an easy fix for this. LTH training requires multiple times of the standard training epochs and longer training would impact the observed results greatly as explained above. There are 3 options for fixing this: (a) Scaling down IMP training such that total number of epochs across different training methods. (b) Scaling up other methods such that they match IMP training (c) Acknowledging this factor in a section at the end and discussing how it can impact different metrics. I.e. accuracy, thus everything else. Flatness. Without this I'm afraid results could be misleading.

- Few related work is missing:
  - [How Well Do Sparse ImageNet Models Transfer?](https://arxiv.org/abs/2111.13445): They study how well pruning solutions transfer.
  - [Gradient Flow in Sparse Neural Networks and How Lottery Tickets Win](https://arxiv.org/abs/2010.03533): They show lottery tickets converge back to a close neighbourhood pruning solutions. Furthermore they have experiments studying the hessian spectrum of the weights.

- I think resnet18 and resnet20 results are a bit redundant. Similarly 3 one-shot pruning methods (synflow, snip, grasp) also correlate highly and given Frankle21's results less interesting to look at. I recommend authors to include Imagenet (instead of one of cifar results)  and RigL/Small-Dense results (instead of two of the one shot pruning methods) in main text.

- It would be nice to indicate matching sparsities for each method/task combination and measure the remaining dimensions for the highest sparsity achievable. One easy way would be to include a vertical line at each plot indicating the highest sparsity achieved by each method without accuracy drop.

# Minor
- "as well as tens of sparsification methods," -> better to give number.
- "find that an appropriate sparsity (e.g., 20% ∼ 99.08%) can yield the winning ticket to perform comparably or even better in all above four " -> 20% sparse is almost dense. I'm not sure this range tell's much. Maybe only using the 99% makes more sense?
- "There were preliminary attempts done in earlier literature" -> Papers cited here cited again in next sentences. It make sense to reduce the repetition, being specific is better as it is done in following sentences.
- "for CIFAR-10 experiments, CIFAR-100 is regarded as the OoD dataset;" -> Is this KNN eval? Not clear how evaluation is done
- What is input flatness? Wasn't clear from the text. Would be nice to define.
- Figure 7: not clear what the dataset and architecture is. Furthermore here two networks with different test accuracies are compared. It would be nice to indicate the test accuracies and if possible compare networks at same test accuracy.

[Frankle21] https://arxiv.org/abs/2009.08576

**Strengths And Weaknesses:**

Strength
- Combining many different evaluation dimensions with extensive set of experiments, which include ResNet-50 on Imagenet.
- Includes dynamic sparse training and small dense baseline.
- Related work is discussed in detail, almost in a survey level.

Weaknesses
- I think authors don't control for total training epochs. Training length could effect many of the dimensions discussed and thus effect the conclusions. For example longer the training is flatter a solution gets. Or longer the training is better the accuracy gets (Imagenet). As seen in the results improved accuracy often improves other metrics.
- 3 one-shot pruning methods included in the main text, while the only dynamic sparse training method is shared in appendix despite being a stronger baseline.
- Though related work discussions are impressive and goes beyond expected, there are few related work I think worth including.

---

> ### Author Response · Authors · 2022-07-27
> **Response to Reviewer jy8z**
>
> Many thanks for acknowledging our extensive & useful experiments and sufficient references. We point-wisely address all your concerns as below:
>
> **[Cons 1. The control of total training epochs.]** As shown in Table 1, for a certain dataset, the number of training epochs for each sparse subnetwork is the **SAME**, since all of them are trained from the same random initialization for a fixed number of epochs (e.g., 182 epochs on CIFAR, and 90 epochs on ImageNet). In other words, the weights in sparse subnetworks from diverse methods are updated with the same number of training epochs, starting from the same initialization.
>
> On the other hand, the number of epochs used for finding sparse masks is different for different methods. To further mitigate reviewer jy8z’s concerns, we follow the provided options and add extra experiments as follows:
>
> - Scaling up SNIP/GraSP/SynFlow such that their used epochs for mask finding match LTH-IMP’s. As shown in Figure A20, LTH-IMP still presents superior performance, compared to IMP with SNIP/GraSP/SynFlow which maintains the number of epochs the same for both mask finding and subnetwork training processes.
> - Examining how the number of epochs during LTH’s mask finding impacts the achievable performance. Figure A21 tells that reducing the number of epochs in mask finding, e.g., only using 10% epochs, leads to moderately degraded performance. Our observations are consistent with the ones in [R1, R2, R3].
>
> **[Cons 2. Missing references.]** Many thanks for pointing out. In our revision, the paper “How Well Do Sparse ImageNet Models Transfer?” will be included in the discussion of sparse transfer in the related work section; the paper “Gradient Flow in Sparse Neural Networks and How Lottery Tickets Win” will be discussed in both the related work section and experiment section of “Geometry of Loss Landscapes”.
>
> **[Cons 3. The content organization.]** Thanks for the constructive comments about the content organization. We will follow reviewer jy8z’s suggestions: (1) moving part of ResNets and one-shot pruning results to the appendix; (2) moving the ImageNet and RigL/Small-Dense results to the main text.
>
> **[Cons 4. Indicate matching sparsities without accuracy drops.]** As presented in Figure 2’s caption, we use solid and dash curves to indicate whether the sparsity is matching or not for each method/task combination. Specifically, the performance curves are divided into two regions: (1) Region I with solid lines, where the sparse subnetworks show matched test accuracy; Region II with dash lines, where the sparse subnetworks have degraded test accuracy. Then, these sparse subnetworks are further evaluated under other metrics, and the drawing style of solid/dash lines is preserved across all figures. We will also add the vertical line at each plot in our final version.
>
> **[Cons 5. Minors.]** Thanks for all the detailed and extremely helpful suggestions. We will accordingly address the minor issues in our final version.
>
> - We will provide a specific number rather than “tens of”.
>
> - In the case of (ResNet-20, CIFAR-100), the LTH-PASS only exists at sparsity levels less than 20%. For the rest cases, the sparsity of LTH-PASS is beyond 50% and approaches 99%. In our revision, we will clarify all the highest sparsity of LTH-PASS in different settings, rather than a sparsity range.
>
> - We will reduce the repetition of citations around “There were preliminary attempts done in earlier literature”, in our final version.
>
> - Following Hendrycks & Gimpel (2016); Hendrycks et al. (2018); Hein et al. (2019); Augustin et al. (2020), for CIFAR-10 experiments, we treat CIFAR-100 as the OoD datasets. During the evaluation, we feed both CIFAR-10 and CIFAR-100 images to the trained model, extracting the largest output logit. If the logit is large than a given threshold, we regard it as an in-distribution sample; otherwise, it is a out-of-distribution sample. Then, we compute the true positive rate (TPR) and false positive rate (FPR). Lastly, we vary the threshold, draw the ROC curve (TPR versus FPR), and calculate the area under the ROC curve (ROC-AUC) which indicates the model’s out-of-distribution generalization ability.
>
> - The input flatness is defined as the Hessian of objective function respect to the input samples. We will add this definition to our revision.
>
> - As shown in the first paragraph of Page 10, the composition neurons in Figure 7 are obtrained from ResNet-50 trained on ImageNet. We will also include it in the caption of Figure 7 in our revision. Actually, this work examies the quality of different sparse connectivities under diverse evaluation metrics, where the sparsity levels are controlled for a fair comparison. For the subnetworks with 59.04% sparsity, LTH and RP have 76.1% and 73.5% test accuracy respectively. If we control the comparison under the same test accuracy of 76.1%, the IoU of composition neurons for LTH:RP is 0.1110:0.1034, and the conclusion remains consistent.

---

> > ### Author Response · Authors · 2022-08-09
> > **Response to Reviewer jy8z [Continued]**
> >
> > [R1] Drawing Early-Bird Tickets: Toward More Efficient Training of Deep Networks
> >
> > [R2] The Sooner The Better: Investigating Structure of Early Winning Lottery Tickets
> >
> > [R3] Efficient lottery ticket finding: Less data is more
> >
> > R.K. Detailed results like figures A20 and A21 are provided in our revised main text and appendix.

---

> ### Comment · Reviewer_jy8z · 2022-08-12
> **Is there a response I should read?**
>
> I haven't seen this any response, is there one?

---

> > ### Author Response · Authors · 2022-08-12
> > **Response to Reviewer jy8z**
> >
> > Dear Reviewer jy8z,
> >
> > Many thanks for asking and so sorry for the inconvenience.
> >
> > We post the response two weeks ago. But it is hid due to the incorrect option of readers, which has been fixed now.
> >
> > Best wishes,
> >
> > Authors

---

> > > ### Comment · Reviewer_jy8z · 2022-08-19
> > > **Thanking the Authors**
> > >
> > > I thank the authors for their response. I read their response and checked the paper/appendix.
> > >
> > > - Vertical lines are quite helpful. It looks like some of the blue lines are missing. Is that on purpose? I.e. you stop at blue vertical line.
> > >
> > > I support acceptance.

---

> > > > ### Author Response · Authors · 2022-08-21
> > > > **Thanking the Reviewer jy8z**
> > > >
> > > > Dear Reviewer jy8z,
> > > >
> > > > Many thanks for supporting the acceptance.
> > > >
> > > > Great catch! Yes, we stop at the blue vertical line on purpose.
> > > >
> > > > As shown in Figure 2, the blue vertical lines indicate the extreme sparsity of LTH subnetworks with comparable clean test accuracy to its dense counterpart.
> > > >
> > > > In later experiments, we only examine sparse subnetworks with competitive clean test accuracy, which lie in the range of (0, the extreme sparsity of LTH subnetworks). And the blue vertical lines overlap the figures' right borders.
> > > >
> > > > Best wishes,
> > > >
> > > > Authors

---

### Review · Reviewer_b8dY · 2022-07-28

**Summary Of Contributions:**

This work conducts a comprehensive assessment of sparse neural networks (with a focus on the lottery ticket method and a few representative methods) from different aspects beyond the test accuracy. Importantly, it shows that there exist winning tickets according to different metrics. Such a careful empirical study is an important contribution to the community and may motivate other researchers to explore the role of sparsity in other performance metrics.

**Broader Impact Concerns:**

I don't see any negative impact.

**Requested Changes:**

See weaknesses. Other than that, there are a few aspects (e.g., transferability, continuous learning and privacy) the authors can potentially look into (not sure if other works have already done that). It has been argued by some researchers that a large and overparameterized network can do better in continuous learning, so I expect sparsity might hurt the performance. For privacy, the added noise is proportional to the dimension, so I expect sparsity could help.

**Strengths And Weaknesses:**

Strengths:
1. The paper is well-written and easy to follow. I really like Figure 1 which clearly illustrates the key take-aways.
2. The authors did very careful experiments on different datasets with different architectures and pruning methods. The results are quite convincing and the experiments are well-designed.
3. Exploring the role of sparsity in other aspects itself is an important contribution. Though some existing papers have explored some of them already, the paper is the first to look at multiple metrics at the same time, which is inspiring.

Weaknesses:
I like this paper. It is a pure empirical paper and the authors didn't overclaim. If I have to nitpick, I think calibration has little to do with uncertainty, so the authors might want to change the name for that section (just call it calibration). In addition, I don't really buy the flatness argument. So to me, the last section doesn't add too much to the paper.

---

> ### Author Response · Authors · 2022-07-28
> **Response to Reviewer b8dY**
>
> Many thanks reviewer b8dY for the very positive assessment and the acknowledgment of our **“well-written”** paper, **“well-designed”** & **“comprehensive”** experiments, **“convincing”** & **“inspiring”** results, and **“important contributions to the community”**. We point-wisely address all your concerns as below:
>
> **[Cons 1. Change the section’s name.]** Thanks for pointing out. We will change the section’s name from “uncertainty” to “calibration” in our final version.
>
> **[Cons 2. Flatness argument.]** We will (1) move the flatness argument to the appendix; (2) move the ImageNet and RigL/Small-Dense results to the main text.
>
> **[Cons 3. More potential aspects.]** Great suggestion!
>
> - For transferability, several existing works [R1, R2, R3] have demonstrated that subnetworks with appropriate sparse connectivities (e.g., from LTH) enjoy undamaged transferability to unseen tasks at around 70% sparsity. We have included the citation and discussion in the introduction & related works.
>
> - For continuous learning, it coincides with reviewer b8dY’s intuition that naively applying pruning to continuous learning hurts the performance. One recent work [R4] proposes a bottom-up lifelong pruning algorithm that successfully identifies the lifelong winning tickets in the scenario of class-incremental continual learning. We have mentioned it in our related work.
>
> - For privacy, [R5] shows the existence of differentially private winning tickets, implying the benefits from appropriate sparsities. It also coincides with reviewer b8dY’s intuition. We will add this discussion in our final version.
>
> [R1] Morcos, Ari, et al. “One ticket to win them all: generalizing lottery ticket initializations across datasets and optimizers.” NeurIPS 2019
>
> [R2] Chen, Tianlong, et al. "The Lottery Ticket Hypothesis for Pre-trained BERT Networks." NeurIPS 2020
>
> [R3] Chen, Tianlong, et al. "The Lottery Tickets Hypothesis for Supervised and Self-supervised Pre-training in Computer Vision Models." CVPR 2021
>
> [R4] Chen, Tianlong, et al. "Long Live the Lottery: The Existence of Winning Tickets in Lifelong Learning." ICLR 2021
>
> [R5] Gondara, Lovedeep, et al. “Training Differentially Private Neural Networks with Lottery Tickets.” ESORICS 2021

---

### Public Comment · ~Surya_Kant_Sahu1 · 2022-07-04
**Relevant prior work not discussed.**

Dear authors, I've read the paper, and thoroughly enjoyed it. Since this work talks about generalization capability of the winning tickets, I would like to bring to your attention a paper we recently released that explores the quality of winning tickets and the difficulty of finding them for different architectures. We find similar results to the results claimed in your work; that excessive pruning retains test accuracy however the expressive power decreases.

We would like you to please read our work and if you believe the results are relevant, please do cite!

Title: "Not All Lotteries Are Made Equal" Link: https://arxiv.org/abs/2206.08175

Thank you.

---

> ### Author Response · Authors · 2022-07-04
> **It's an interesting, yet concurrent work.**
>
> Hi Surya,
>
> Thanks for bringing your recent work up. We will read with interest.
>
> However, your work was submitted to arXiv on Jun 16, and ours was submitted to TMLR in June 29 - less than two weeks after (and our draft was obviously completed way earlier than this time). Therefore, based on the commonly accepted standard of any top-tier ML/CV/NLP conference, the two papers fall under the "concurrent work" case, and there is no obligation for either one to cite/compare with the other.
>
> We would be willing to cite your work out of grace and respect. We can potentially discuss it in a later version of our work if it's found
> by us as indeed relevant.
>
> But we will FIRMLY NOT accept being accused of "relevant prior work not discussed" (which is a serious accusation that we don't seem to deserve at all). We politely request you to consider respect and grace too.

---

> > ### Public Comment · ~Surya_Kant_Sahu1 · 2022-07-05
> > **Clarification**
> >
> > Dear Authors,
> >
> > I apologise for the poorly-worded title of the comment.
> > I would like to clarify that the intention of the comment was not to accuse you of anything, instead it was to bring attention to our work.
> >
> > On a side note, our work has been public since January in openreview: https://openreview.net/forum?id=ugXMIHeasoz
> >
> > Best,
> > Surya

---

> > > ### Author Response · Authors · 2022-08-09
> > > **The reference is cited.**
> > >
> > > Hi Surya,
> > >
> > > The paper "Not All Lotteries Are Made Equal" has been cited and discussed in our updated related work, which is marked by blue.
> > >
> > > Thanks!
> > >
> > > Best wishes,
> > >
> > > Authors

---

### Author Response · Authors · 2022-08-09
**General Response**

Dear AE and all reviewers,

We sincerely appreciate all reviewers’ and AEs’ time and efforts in reviewing our paper. We truly thank all for the insightful and constructive suggestions, which helped further improve our paper.

Most of the promised changes (>99%) and newly added experiments have been included in our revised main draft and appendix, which are marked in blue. We will keep updating our draft.

We are confident that our response should have cleared the air, and we are happy to answer any additional questions and provide more information.

We really thank all reviewers’ and AEs' time and efforts again.

Best wishes,

Authors

---

### Decision · Action_Editors · 2022-09-12

**Recommendation:** Accept as is

**Comment:**

Thank you the authors and reviewers for great discussions. Since the authors have implemented the requested changes and addressed the concerns, all reviewers are leaning to acceptance and I vote for acceptance of the paper. Dear Authors please upload the camera ready version.

Best regards, Action Editor